# Transient polymorphisms in parental care strategies drive divergence of sex roles

Xiaoyan Long [1,2] & Franz J. Weissing [1] ✉

The parental roles of males and females differ considerably between and within species. By means of individual-based evolutionary simulations, we strive to explain this diversity. We show that the conflict between the sexes creates a sex bias (towards maternal or paternal care), even if the two sexes are initially identical. When including sexual selection, there are two outcomes: either female mate choice and maternal care or no mate choice and paternal care. Interestingly, the care pattern drives sexual selection and not vice versa. Longer-term simulations exhibit rapid switches between alternative parental care patterns, even in constant environments. Hence, the evolutionary lability of sex roles observed in phylogenetic studies is not necessarily caused by external changes. Overall, our findings are in striking contrast to the predictions of mathematical models. We show that the discrepancies are caused by transient within-sex polymorphisms in parental strategies, a factor largely neglected in current sex-role theory.

In the animal kingdom, species differ remarkably in the way and degree female and male parents are involved in parental care[1,2]. In virtually all mammals, most of the care is provided by females[1,3], while in birds, biparental care is the most prevalent pattern[1,4]. In teleost fishes, male-only or male-biased care occurs regularly, next to a variety of other care patterns[1,5]. Even within species, parental care patterns can be highly diverse[6]. For example, in Eurasian penduline tits (*Remiz pendulinus*), female-only care and male-only care co-occur in the same population[7], while in Chinese penduline tits (*Remiz consobrinus*), female-only care, male-only care, and biparental care all coexist[8]. Moreover, phylogenetic studies suggest that parental care patterns are highly dynamic in that transitions between patterns occur frequently[9–11].

The explanations that have been proposed for sex differences in parental roles often initiated heated debates in the literature. One debate centres around the role of anisogamy (the difference in gamete size between males and females). Robert Trivers[12] argued that anisogamy explains why in many taxa females tend to invest more in post-zygotic parental care than males. According to Trivers, females risk losing a larger initial investment in the ovum if they abandon the clutch. Some authors criticised Trivers' argument, stating that optimal decision-making should be based on future costs and benefits, not past

investment[13]. While agreeing with this critique, other authors pointed out that Trivers' prediction can be revived when other factors are considered[14,15]. This viewpoint is, in turn, hotly debated[16–19]. Another debate in the literature is on whether and how the relative abundance of males and females drives parental sex roles[20]. A popular theory predicts that the 'operational sex ratio' should play a decisive role because the overrepresented sex on the mating market should be predestined for taking on the parental care tasks[21]. More recently, attention has shifted to the 'adult sex ratio' as a predictor of sex differences in parental sex roles[22–25]. Last but not least, there is debate in the literature on the role of sexual selection in determining parental sex roles[12,14,26]. All these debates are intricate in themselves; moreover, they are interwoven because initial investments, sex ratios, and sexual selection are mutually dependent.

In a situation like this, where the outcome of evolution is determined by the interplay of mutually dependent factors, verbal theories can easily lead astray. As a significant step forward, Kokko and Jennions[22] developed a comprehensive modelling framework, allowing them to disentangle the role of the various factors involved in the evolution of parental sex roles. In this framework, male and female fitness functions are derived from a scheme that describes the interactions of the sexes in a population. These functions are then analysed

[1]Groningen Institute for Evolutionary Life Sciences, University of Groningen, Groningen 9747AG, The Netherlands. [2]Present address: Institute of Biology I, University of Freiburg, Freiburg im Breisgau 79104, Germany. ✉e-mail: f.j.weissing@rug.nl

mathematically (see 'Methods'), allowing to predict how sex differences in life-history parameters, biased sex ratios, multiple mating, and sexual selection affect the evolution of parental sex roles. However, to keep the model analytically tractable, the factors involved must be stripped to their bare-bone essentials. For example, the dynamic process of sexual selection is reduced to a set of fixed parameters that cannot coevolve with parental strategies. Moreover, the calculations are not trivial and are error-prone. Indeed, Fromhage and Jennions[27] pointed out mistakes and erroneous conclusions in the study of Kokko and Jennions[22].

For these reasons, we here re-evaluate some predictions of sex role theory by means of a simulation approach that is based on (an extended version of) the modelling framework of Kokko and Jennions[22]. Individual-based evolutionary simulations lack the rigour of mathematical analysis, but they have several advantages[28]. First, more natural assumptions can be made concerning the inclusion of sexual selection or factors such as sex differences in pre-mating investment. Second, a simulation approach does not require the (error-prone) derivation of fitness functions, and it can avoid analytical short-cuts (such as the assumption that evolutionary trajectories follow the steepest ascent of the fitness function). Third, individual variation emerges in a natural way, making it possible to study its evolutionary implications[29–31].

Here, we show that there is an intrinsic evolutionary tendency to create asymmetry in parental care between the two sexes. Even in the absence of sex differences and starting with egalitarian biparental care, evolution invariably results in either strongly female- or male-biased care. In addition, we show that in the absence of external change, a population can rapidly switch from one care pattern to the other over evolutionary time. This provides an alternative explanation for the evolutionary lability of parental sex roles observed in phylogenetic studies, which is currently assumed to be caused by environmental change. Moreover, our study suggests that, contrary to common belief, sexual selection is not the driving force behind parental sex roles. In contrast, we show that parental sex roles evolve first and subsequently initiate mate choice and mate competition. Our simulation outcomes are surprisingly different from earlier mathematical predictions[27], despite the fact that very similar assumptions are made. The reason for this is that the main analytical approaches (e.g., selection gradient methods) used to derive results are based on the assumptions of a monomorphic population, whereas our simulations show that parental conflict drives both male and female populations to

a polymorphic state. The polymorphic state in our simulations is only transient, as it disappears as soon as sex role differentiation has occurred. Yet, they leave a lasting mark on the course and outcome of evolution.

## Results

In a nutshell, our model (Fig. 1a, see 'Methods' for details) follows individual males and females from birth to death. After maturation, adult individuals can be in one of two states: the mate search state and the caring state. Individuals seek mating opportunities in the mate search state; once mated, both members of the mated pair switch to the caring state. Each individual provides care for a time period corresponding to its inherited sex-specific parental care strategy and switches back to the mate search state afterwards. The total amount of care provided by both parents determines the survival probability of the offspring in the clutch (Fig. 1b). The offspring inherit the care strategies from their parents (according to Mendelian inheritance and subject to rare mutations of small effect size). Parental care strategies must strike a balance between caring as efficiently as possible and mating as often as possible. Both caring and mate search are costly since individuals can die in any state, with a mortality rate that depends on their state and sex. Strategies that perform well are transmitted to a large number of offspring, thereby increasing in relative frequency in the population. Over the generations, an evolutionary equilibrium emerges during the simulation; fitness calculations are not required for this. As explained in 'Methods', the model can easily be extended to include sexual selection and sex differences in pre-mating investment.

Although the model is very similar in set-up and spirit to the analytical models mentioned above, we will now show that the evolutionary outcome is remarkably different from that reported in the earlier studies of parental sex role evolution.

### Sex-biased care evolves in the absence of sex differences

First, we consider the scenario where mating is at random and the sexes do not differ in their life-history parameters (Fig. 2). Based on the analytical model, Kokko and Jennions[22] predicted the evolution of egalitarian biparental care for this scenario. Correcting a mistake in the fitness calculations, Fromhage and Jennions[27] showed that instead the analytical model predicts convergence to a line of equilibria. If we apply the selection gradient method of refs. 22 and 27 to our slightly modified model, we arrive at the same conclusion (Fig. 2a): the care effort of females and males converges to an

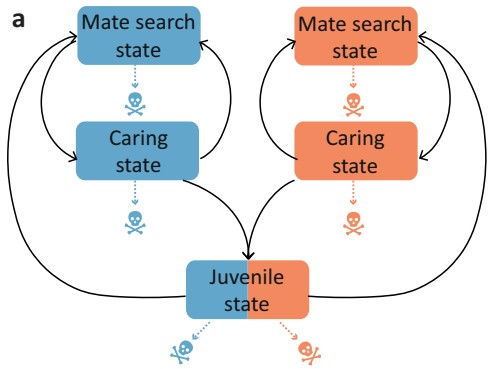
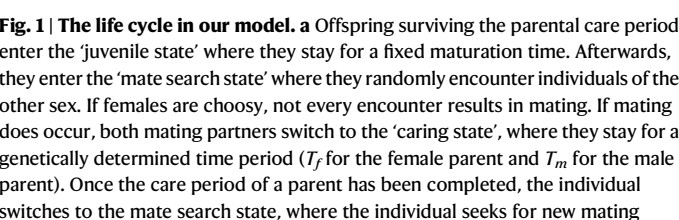
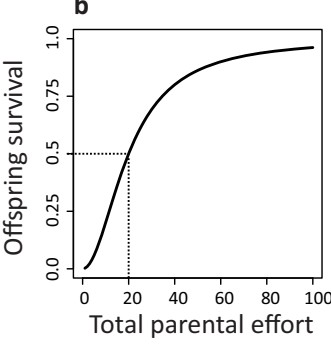

**Fig. 1 | The life cycle in our model. a** Offspring surviving the parental care period enter the 'juvenile state' where they stay for a fixed maturation time. Afterwards, they enter the 'mate search state' where they randomly encounter individuals of the other sex. If females are choosy, not every encounter results in mating. If mating does occur, both mating partners switch to the 'caring state', where they stay for a genetically determined time period ($T_f$ for the female parent and $T_m$ for the male parent). Once the care period of a parent has been completed, the individual switches to the mate search state, where the individual seeks for new mating

opportunities. In all states, mortality can occur. In the random mating scenario, individual life expectancy is 1000 time units (= 'days'). For simplicity, we equate this time period with one 'generation'. **b** Offspring survival is proportional to $S(T_{tot}) = T_{tot}^2/(T_{tot}^2 + B^2)$, an increasing sigmoidal function of total parental care. In the case of uniparental care, the optimal care duration is equal to $B$ (see 'Methods'), which is a useful benchmark expectation. Throughout, we consider $B=20$. Colour conventions: throughout the manuscript: females are indicated by the colour red, and males are indicated by the colour blue.

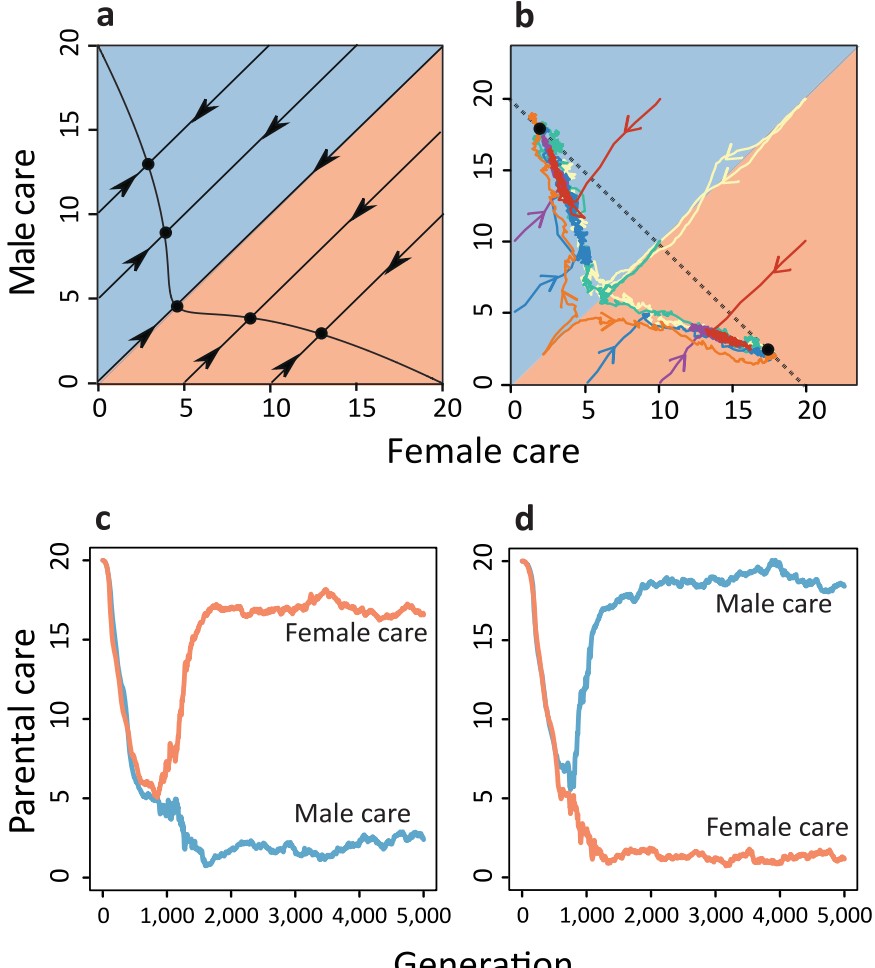

**Fig. 2 | Evolution of sex-biased parental roles in the absence of sexual selection.** The graphs depict evolutionary trajectories when mating is at random and males and females do not differ in their life-history parameters. **a** For this scenario, the selection gradient method predicts convergence to a curve of equilibria (solid black line). **b** In contrast, individual-based simulations converge to one of two equilibria (black dots) corresponding to either strongly female-biased care or strongly male-biased care. The black dotted line in (**b**) corresponds to those care levels where the sum of female and male care equals the benchmark value $B=20$.

Differently coloured lines in (**b**) indicate different initial conditions. Replicate simulations starting with egalitarian care levels converge, with equal probability, to **c** the female care equilibrium or **d** the male care equilibrium. The time trajectories shown in (**c**, **d**) correspond to the two simulations depicted as yellow-coloured lines in (**b**). The red and blue lines in (**c**, **d**) depict the average levels of female and male care in the evolving population. Population sizes fluctuated around 2000 females and 2000 males.

equilibrium; there is a continuum of equilibria, which are located on a curve that includes a broad spectrum of parental care patterns. In other words, it is possible to obtain any type of care strategy, including female-only care, egalitarian biparental care, male-only care, and everything in between. The evolutionary outcome is fully determined by the initial conditions.

In contrast to these analytical predictions, our individual-based simulations never resulted in egalitarian care or a line (or curve) of equilibria. Instead, all our simulations (>10,000 generations for different parameter values and different initial conditions) converged to one of two stable equilibria corresponding to either strongly female-biased care or strongly male-biased care. Initial conditions with sex-biased care tended to converge to the corresponding sex-biased equilibrium, while initial conditions without sex bias converged to each of the two equilibria with equal probability (Fig. 2b). Figure 2c, d shows the time trajectories of two replicate simulations starting at a high level of egalitarian care. In a first phase, both populations follow the analytical prediction and converge to a low level of egalitarian care. Then strongly sex-biased care evolves, along the curve of equilibria of the analytical model. Both stable equilibria have the property that the total care provided by the two parents equals $B=20$.

### The evolution of sex-biased parental roles is driven by transient polymorphisms

Figure 3 shows that the tendency towards sex-biased care is not an artefact of the simulation model, but rather a result of sexual conflict generating polymorphisms, which in turn lead to the evolution of strongly male- or female-biased care. In the simulation shown, the population was initialised at the same care level ($T_f = T_m = 20$) for females and males. Hence, initially, the sum of the parental care levels exceeds the value $B = 20$. Accordingly (see 'Methods'), there is a strong selection in both sexes to reduce the level of care. In the first 800 generations, the care levels in males and females rapidly decline until a value of 5 is reached in both sexes (Fig. 3a, b), in line with the predictions of the selection gradient approach (see Fig. 2a). At this care level, the mortality of offspring is very high, and additional care would provide a considerable benefit. Yet, the parents are caught in a cooperation dilemma: both are interested in the survival of their offspring, but each parent is better off if most of the care is provided by the other parent[12,32,33]. The U-shaped fitness profiles in Fig. 3c show that the two contrasting selection pressures result in disruptive selection, where extreme strategies have the highest fitness. In other words, individuals that provide a relatively high level of care would benefit from

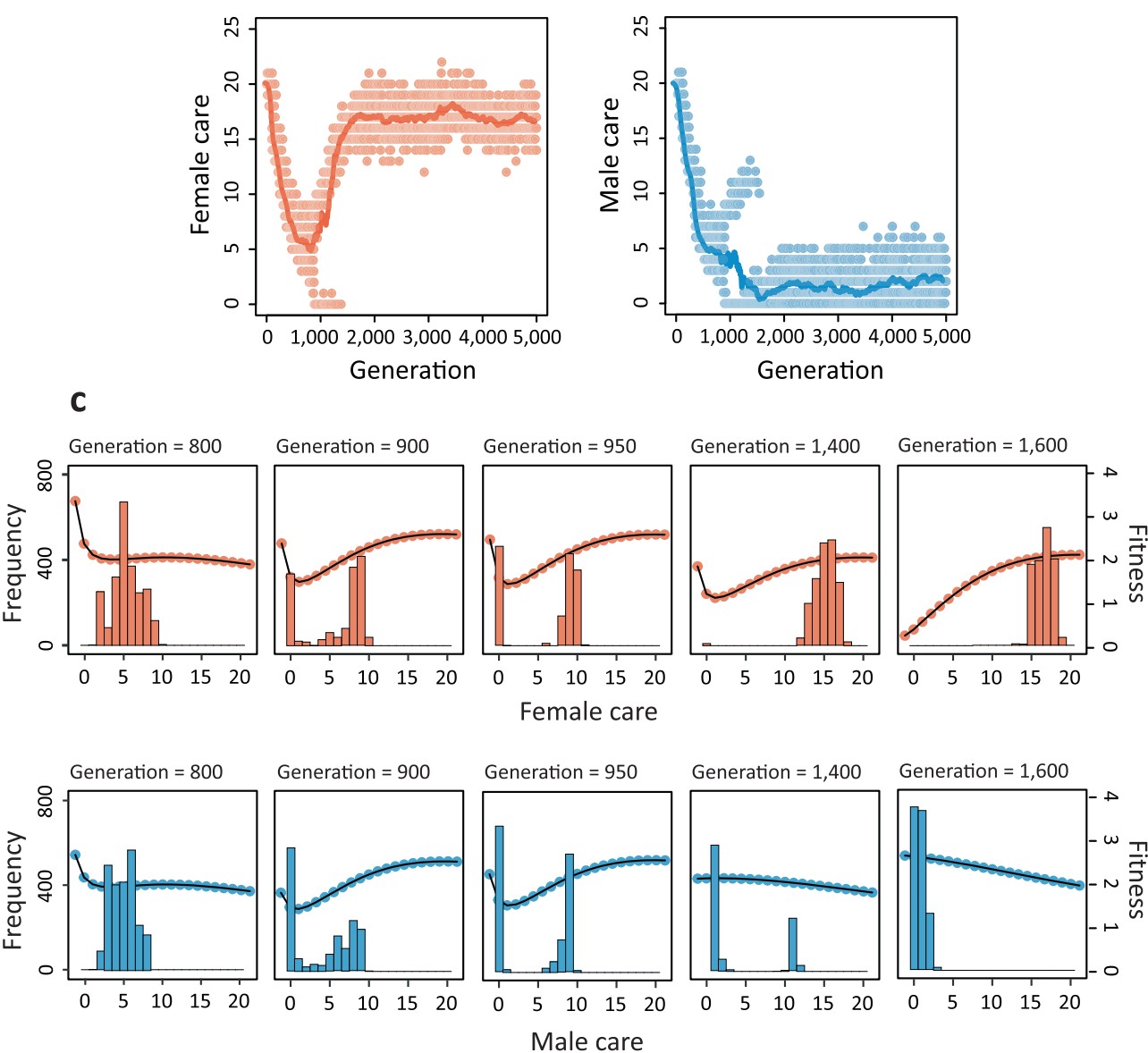

**Fig. 3 | Sex role divergence is driven by transient polymorphisms in both sexes.** Evolution of **a** female and **b** male care for the simulation in Fig. 2c. Lines show the average care levels of females (red) and males (blue) in the population, while dots represent individual care levels. **c** For five different generations, the histograms (left axis) show the distribution of care levels in females (red) and males (blue). The fitness profiles (right axis) indicate in each case the expected lifetime reproductive success of females and males with care strategies ranging from 0 to 20 in the corresponding population.

providing even more care, while those that provide relatively little care would benefit from caring even less. In generations 900–950, disruptive selection has led to a bimodal distribution of care strategies in each sex. This indicates that there are two types of females and two types of males: one type not caring at all and the other type caring at a level around 10. Such a population is not very efficient, because many matings result in no care at all or a low care level of around 10. Sex role differentiation provides an escape route from this unfortunate state of affairs[34]: one of the two care patterns becomes associated with the female sex, while the other becomes associated with the male sex. As a result, the within-sex polymorphisms in care strategies only last for a short period. In the simulation of Fig. 3, the high-care strategy becomes associated with the female sex and the no-care strategy becomes associated with the male (the opposite happened in 50% of the simulations): in generation 1400, the no-care strategy has almost disappeared in females and selection is directional in males (in favour of the no-care strategy). In the end (generation 1600), directional

selection keeps the care level low in males, while stabilising selection keeps the care level just below 20 in females. Without exception, the same sequence of events (with similar timing) was observed in thousands of simulations starting with similar care levels in the two sexes: the population first converges to the state $T_f = T_m = 5$, where a transient polymorphism emerges that eventually results in pronounced parental sex roles.

The above considerations are confirmed by mathematical analysis using an adaptive dynamics approach (Supplementary Fig. 2). The analysis reveals that there is indeed first directional selection towards lower care level until an 'evolutionary branching point'[35] is reached at a care level of 5. Here, directional selection turns into disruptive selection, which produces a polymorphism of care strategies. In order to apply an adaptive dynamics approach, we had to assume that the care level of individuals is independent of their sex. Accordingly, this approach cannot predict the evolutionary emergence of sex roles (see ref. 34).

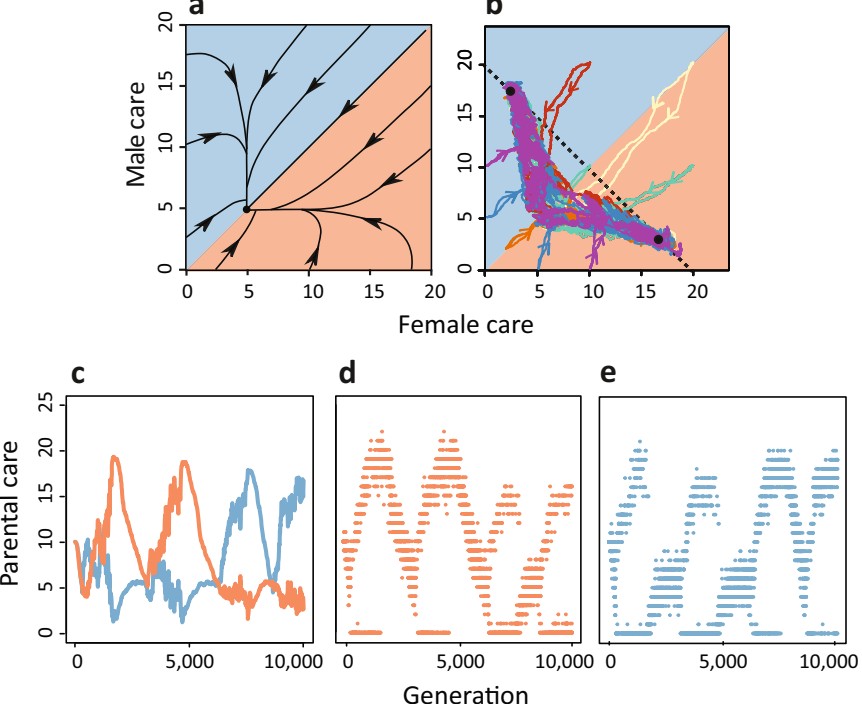

**Fig. 4 | Evolution of parental roles when biparental care has a synergistic effect.** In contrast to Fig. 2, where the two parents have an additive effect on offspring survival ($\sigma = 0$), we here consider the case of biparental synergy ($\sigma = 0.05$). **a** Now the analytical sex role model (ref. 27) predicts the evolution of egalitarian biparental care (black dot). **b** The individual-based simulations still have a strong tendency towards sex role differentiation: most of the time, the simulations are close to the two black dots, representing strongly male-biased care and strongly female-biased care. However, the evolutionary trajectories repeatedly switch between these two care patterns. Differently coloured lines in (**b**) indicate different initial conditions. **c** This representative simulation shows the relatively rapid (in evolutionary time) succession of strongly male-biased and strongly female-biased care. Throughout, there is considerable variation in (**d**) female and **e** male care strategies. Notice that changes in the 'direction' of evolution are always associated with extreme transient polymorphisms in both sexes, where the no-care strategy coexists with a high-care strategy. Lines in (**c**) show the average care levels of females (red) and males (blue) in the population, while dots in (**d**, **e**) represent individual care levels.

We also ran simulations with sex-biased initial care levels. As indicated in Fig. 2b, simulations with an initially relatively small bias in, say, female-biased care typically ended up in the strongly female-biased care equilibrium. This can happen in two ways. If a simulation already starts in the vicinity of a strongly sex-biased care equilibrium, it will converge to that equilibrium without undergoing a period of transient polymorphisms. Otherwise, the simulation will first converge to the 'branching point' $T_f = T_m = 5$, where again polymorphisms emerge in both sexes, eventually leading to strong sex-biased care. However, now the two possible outcomes are no longer equally likely. As illustrated and explained in Supplementary Fig. 3, the initial sex bias in parental care leaves its mark on the polymorphisms and the resulting equilibrium: both outcomes are possible, but it is much more likely that the initial sex bias is enhanced than that it is reversed.

## Biparental synergy can lead to fluctuating polymorphism or inefficient biparental care

In contrast to the simulations reported above, egalitarian biparental care occurs in many bird and fish species, and in other animal taxa[1–4]. A potential reason is that in natural populations the parents complement each other, thereby providing more benefits to their offspring than the sum of their individual contributions[36–38]. Division of labour or other sources of synergy among the parents could reduce sexual conflict about who should do the caring and strongly select for biparental care[33,39]. Here, we introduce parental synergy in our model in line with earlier modelling studies[27,40]: we assume that the care levels $T_f$ and $T_m$ of the two parents provide a benefit $T_f + T_m + \sigma T_f T_m$ to their offspring, where the degree of synergy $\sigma$ is a positive parameter (in the additive model considered until now, $\sigma = 0$). In the analytical model of

Fromhage and Jennions[27], the introduction of a small degree of synergy transforms their line of equilibria into a single stable equilibrium corresponding to egalitarian biparental care.

Figure 4 considers the case of relatively weak synergy ($\sigma = 0.05$). As shown in Fig. 4a, the selection gradient approach indeed predicts the evolution of egalitarian biparental care, irrespective of the initial conditions. Again, the individual-based simulations (Fig. 4b) differ strikingly from this prediction. As in Fig. 2b, all simulations converged to either strongly female-biased care or strongly male-biased care. However, as illustrated by a representative simulation in Fig. 4c, the average care levels in both sexes exhibit large fluctuations, corresponding to rapid transitions between female-biased and male-biased care. Moreover, most of the time, there are considerable polymorphisms in the care levels of both sexes (Fig. 4d, e), and once in a while, there are brief periods of egalitarian care (where the average care levels of both parents are very similar). Whenever such a situation arises, a similar phenomenon occurs as in Fig. 3. First, both sexes become strongly polymorphic for a no-care strategy and a high-care strategy, but this polymorphism is transient and breaks down, giving way to the re-establishment of strongly female-biased or strongly male-biased care.

In the case of a larger degree of synergy ($\sigma = 0.20$), the population converges to egalitarian care (Supplementary Fig. 4a), although both the female (Supplementary Fig. 4a$_2$) and the male population (Supplementary Fig. 4a$_3$) remain highly polymorphic. The average care levels (Supplementary Fig. 4a$_1$) in both sexes are about $T_f = T_m = 5$ and, hence, very low. Taking synergy into account, this investment results in a total care level of about $5 + 5 + 0.2 \cdot 25 = 15$. This is considerably less than in the additive model without synergy (Fig. 2b), where in both

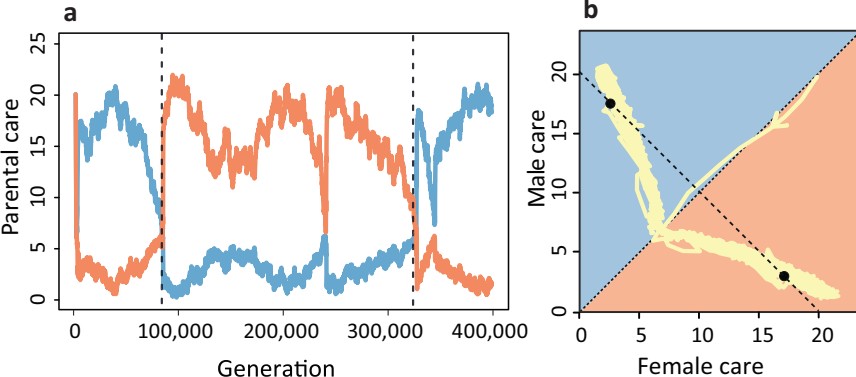

**Fig. 5 | Evolutionary lability of parental sex roles.** When simulations were run for extended periods, transitions occurred between the two stable equilibria. In other words, long periods of male- or female-biased care were followed by rapid switches to a situation where most of the care was provided by the other sex. Here, this is shown for a long-term simulation of the random mating scenario in Fig. 2, but with a 1-day pre-mating period in both sexes (see 'Methods'). The time trajectory in (**a**) shows a rapid switch from male-biased care to female-biased around generation 90,000, and a rapid switch from female-biased care to male-biased care around generation 320,000. The red and blue lines depict the average levels of female and male care in the population. In (**b**), the same long-term trajectory is shown in trait space. This plot indicates that the switches between the alternative equilibria (black dots) occur in a characteristic manner, with a trajectory that follows the curve of equilibria of the analytical model shown in Fig. 2a.

non-egalitarian equilibria the total care level equals $B = 20$. Apparently, the introduction of synergy does not allow the parents to escape from the cooperation dilemma through the evolution of either male-biased or female-biased care. Instead, the conflict between the sexes continues, resulting in a broad spectrum of care strategies and an outcome that is, regarding offspring survival, relatively inefficient. This conclusion only changes for a very high degree of synergy ($\sigma = 2.0$, Supplementary Fig. 4b): now the population converges to an egalitarian care level satisfying $T_f + T_m + \sigma T_f T_m = B$.

### Evolutionary lability of parental sex roles
The switches between two alternative equilibria that we observed in Fig. 4b are not restricted to the case of (weak) parental synergy. They also occur regularly in the absence of synergy ($\sigma = 0$), but on a much longer time scale. This is demonstrated in Fig. 5, which shows that, on a long-term perspective, rapid switches from one equilibrium to the other occur regularly. Accordingly, our simulations suggest that parental roles can be evolutionarily labile. This is in line with phylogenetic studies, which also conclude that parental care patterns are highly dynamic and that, on a long-term perspective, transitions between different care patterns have occurred frequently in many animal taxa[9–11].

The average time between switches depends on the degree of stochasticity and the strength of attraction, which in our case corresponds to population size and the steepness of the selection gradients. Decreasing the population size by relaxing density dependence or by increasing the mortality rate for both sexes did indeed lead to much faster transitions between states (Supplementary Figs. 5 and 6). The same happened when we weakened selection by prolonging the pre-mating period (see 'Methods') in one or both sexes (as in Fig. 5).

After analysing many transitions between sex roles, we observed the following commonalities (see Supplementary Fig. 7). Although the two strongly asymmetric equilibria $(T_f, T_m) = (17.5, 2.5)$ and $(2.5, 17.5)$ are stable, the care levels of both sexes fluctuate to a certain extent (see Fig. 5a and Supplementary Fig. 7), indicating that stabilising selection is weak. As soon as, during these fluctuations, the care level of the less-caring sex approaches 5, the care level of the other sex also rapidly drops to 5. In other words, the system converges to the evolutionary branching point $T_f = T_m = 5$ where a transient polymorphism emerges, which can either result in the old equilibrium (as happens in generation 315,000 in Supplementary Fig. 7) or in the alternative equilibrium (as happens in generations 220,000 in Supplementary Fig. 7).

### Joint evolution of mating and parental strategies
Mating and parental care strategies are closely interrelated, but the causal relationships between the two types of strategy are difficult to disentangle. Mathematical models incorporating both factors tend to be analytically intractable and can only be solved by iteration methods[40]. Many models on the evolution of parental roles therefore represent mating patterns by a parameter that cannot change in time (e.g., the 'strength of sexual selection' in refs. 22 and 27). It is a clear advantage of individual-based simulation models that various scenarios for the joint evolution of mating and parental care strategies can be implemented in a natural way. To demonstrate this, we extended the baseline version of the model by allowing female preferences and male ornaments to evolve alongside the parental strategies. In our Fisherian model[41] (also called the 'sexy-son model'), female preferences and male ornaments are characterised by heritable parameters $p$ and $s$, respectively. When female preferences are zero, all males have the same probability of being chosen, and mating occurs at random. When female preferences are above zero, males with large ornaments are preferred. Male ornamentation is costly in that it negatively affects male survival. Female choosiness is costly, because choosy females may take a longer time before they find a mate.

Figure 6a shows some representative simulations, all starting with random mating ($p = s = 0$), but with different initial levels of parental care. All simulations converge to one of two equilibria that are characterised by either male-biased care or female-biased care. Whenever male-biased care evolved (Fig. 6b), female preferences stayed at a very low level, corresponding to random mating. Whenever female-biased care evolved (Fig. 6c), female preferences for male ornaments evolved as well, together with elaborate male ornamentation. In all simulations leading to female-biased care, female choosiness only got off the ground after female care level had reached relatively high level.

Again, these two types of equilibrium do not persist forever. As shown in Supplementary Fig. 8, each equilibrium defines the dominant sex role pattern for long periods of time (many thousands of generations), followed by a rapid switch to the other type of equilibrium. These transitions proceed in both directions. We investigated many of these transitions, and in all cases the parental strategy changed first (either from male-biased care to female-biased care or vice versa), followed by the emergence or disappearance of female choosiness and male ornamentation. From this, we tentatively conclude that, at least for the mating strategies considered in our simple model, the causal relationship goes from parental sex roles to mating roles, and not the other way around.

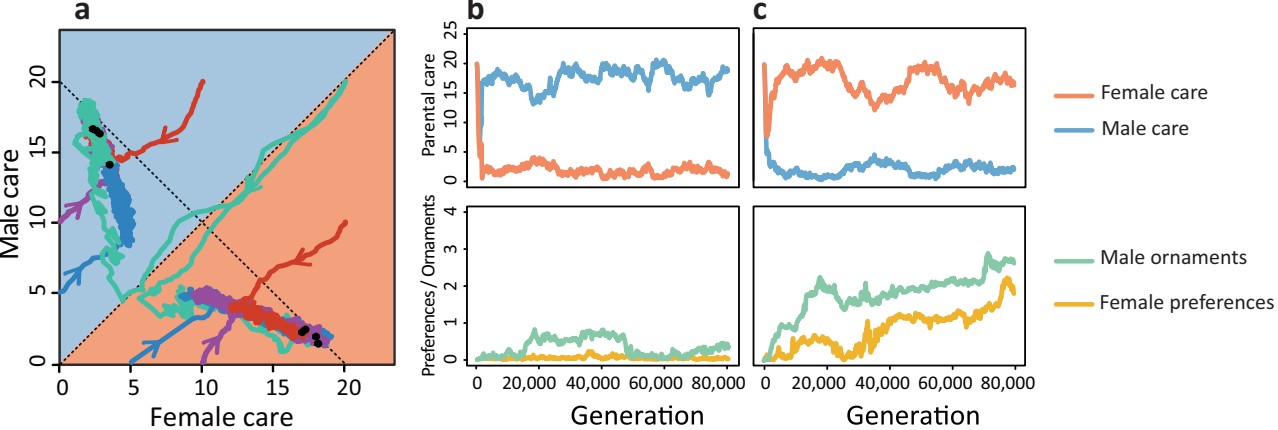

**Fig. 6 | Joint evolution of mating and parental strategies. a** If parental care strategies evolve alongside the evolution of female preferences for a costly male ornament, all simulations result in one of two alternative equilibria. Differently coloured lines in (**a**) indicate different initial conditions. **b** One equilibrium is characterised by male-biased care, the absence of female preferences, and a small degree of male ornamentation. **c** The other equilibrium is characterised by female-biased care, strong female preferences, and a high degree of male ornamentation. In this simulation, there was no pre-mating period and no parental synergy. Lines in (**b**, **c**) show the average female care level (red), male care level (blue), female preferences (yellow) and male ornaments (green) in the population.

## Asymmetry in pre-mating investment affects the evolution of parental sex roles

In most taxa, females tend to invest more in post-zygotic parental care than males[1–4]. Since females are, by definition, the sex-producing larger gametes, it is plausible to assume that anisogamy plays an essential role in the evolution of parental sex roles[12]. Trivers' argument that the sex with the highest pre-mating investment is predestined to invest more in post-zygotic care because it has 'more to lose' is generally considered to be flawed[13], but various authors pointed out other causal links from anisogamy to female-biased care, via secondary effects of anisogamy, such as higher competition among males or a lower certainty of parentage in males[14,15]. To investigate the role of pre-mating investment, we extended our model by introducing a pre-mating period for one of the sexes (see Supplementary Fig. 1). After any parental care period, an individual of that sex has to spend a fixed number of days with other activities (like growing a new clutch of eggs in females or building a new nest in males) before entering the mate search phase again (see 'Methods' for details). Mating is still assumed to be at random, and there are no other differences between the sexes.

Figure 7 shows, for four mortality levels in the pre-mating period, that the sex with the higher pre-mating investment tends to evolve a higher degree of post-zygotic parental care in most cases. This trend is very pronounced if the mortality in the pre-mating period is five times as high as in the mate search period (black curve). This is not too surprising: the sex with higher mortality has a shorter life expectancy; this, in turn, makes every mating very valuable, shifting the balance between current and future reproduction towards a higher investment in the current clutch[42,43]. However, this cannot be the whole story, as the 'Trivers effect' is also noticeable when the pre-mating period does not affect life expectancy at all (white curve: zero mortality in the pre-mating state). We initially thought[44] that this outcome results from the fact that the sex with the shorter pre-mating period has a higher variance in mating success, which selects for higher mating effort and reduced parental care[45]. However, our simulation data and mathematical analyses do not support this explanation. Based on the analysis of many simulations, we now think that the outcome results from the fact that the sex with the shorter pre-mating period spends more time in the mate search state (waiting for mates that are still in the pre-mating period), and that, accordingly, the mortality of that sex is higher (as the mortality is higher in the mate search state than in the zero-mortality pre-mating state). The sex with lower mortality is overrepresented in the adult population. According to the 'Fisher condition'[27,31,46], the

members of the lower-mortality sex have a lower per capita reproductive output, which in our model corresponds to a lower expected number of future matings. In the trade-off between current and future reproduction, the overrepresented sex (here: the sex with the longer pre-mating period) should therefore invest more in the current offspring, which is in line with the white curve in Fig. 7. The reader should notice that the 'life-history argument' for the explanation of the black curve (the sex with higher mortality has fewer expected matings and, hence, should invest more in the current brood) and the 'sex ratio argument' (based on the Fisher condition) for the explanation of the white curve (the sex with lower mortality has a lower per capita reproductive output and, hence, should invest more in the current brood) lead to contrasting predictions on the relationship between sex-specific mortality and parental investment.

In ref. 44, we also extrapolated our findings for extremely high (black curve) and extremely low (white curve) mortality to conclude that "Trivers was right, be it for the wrong reason". Figure 7 shows that, actually, the situation is more complicated if the mortality costs are at an intermediate level (light grey and dark grey curves). In these cases, Trivers' prediction that the sex with the higher pre-mating investment is more likely to evolve a higher degree of post-zygotic parental care only holds in those cases where the sex bias in the pre-mating period is relatively large. This may be explained by the above 'life-history argument': the sex with a (long) pre-mating period has a lower life expectancy, making each clutch more 'valuable' than for the other sex. If, however, the sex differences in duration of the pre-mating period are small, this argument is apparently not decisive, as in the majority of simulations the sex with the higher pre-mating investment ended up in the equilibrium with lower post-zygotic care. It is possible that in these cases the above 'sex ratio argument' is more important. However, in a situation like this, where different lines of reasoning lead to contrasting conclusions, verbal arguments are insufficient to unravel the 'web of causation'. We therefore conclude that we do not have a fully convincing explanation for the pattern generated by our simulations. A reviewer of a previous version of this article hypothesised that sex differences in the approach to the branching point might explain the observed pattern, but this does not seem to be the case, as the same pattern as in Fig. 7 emerges when the simulations are initialised at $T_f = T_m = 5$ (see Supplementary Fig. 9a). In contrast, asymmetric starting conditions have a clear effect on the outcome. As illustrated in Supplementary Fig. 9b, a strong asymmetry in pre-zygotic investment is required to overcome the general tendency that the sex that initially

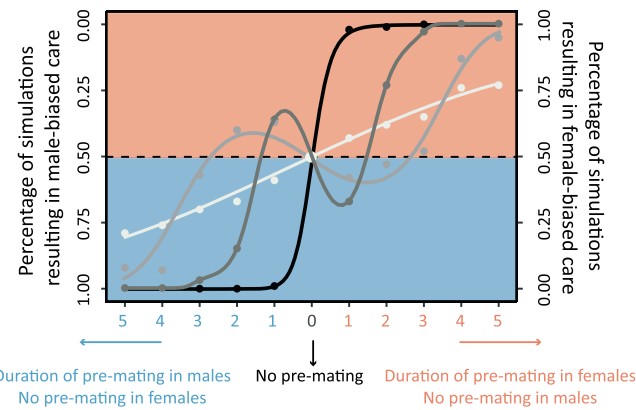

**Fig. 7 | Effect of sex-biased pre-mating investment on parental sex roles.** Percentage of simulations resulting in male-biased care (left axis) or female-based care (right axis) depending on the duration of the pre-mating period in either males (blue) or females (red). Mortality in the pre-mating period was (1) zero (white dots and white line fitted by logistic regression); (2) 0.001, the same as in the mate search phase (light grey dots and line); (3) 0.002, twice as high as in the mate search phase (dark grey dots and line); (4) 0.005, five times as high as in the mate search phase (black dots and line). 100 replicate simulations with 100,000 generations were run per parameter setting, all starting from egalitarian care ($T_f = T_m = 20$). All of these 4400 simulations resulted either in female-biased care or male-biased care.

cares more is more likely to be the caring sex in the end. Interestingly, also in this case the light grey curve is not monotonically increasing, indicating that the causal link between pre-mating investment and post-zygotic parental care is more complicated than Trivers and others (including ourselves) envisaged.

## Discussion

Here we investigated an individual-based simulation implementation of a modelling framework[22] that may be viewed as the cornerstone of sex role evolution theory. Although we made very similar assumptions, we arrived at remarkably different conclusions than the earlier mathematical analyses. First, the populations in our 'null model' (random mating, no sex differences in life-history parameters) do not evolve to egalitarian care[22] or to a line (or curve) of equilibria[27] but rather to one of two stable equilibria corresponding to strongly male-biased or strongly female-biased care, respectively. Second, parental synergy does not necessarily lead to egalitarian care. Even if it does, the evolutionary outcome is not necessarily efficient. Third, our simulations reveal that, as in the analytical models[22,27], sexual selection can lead to a situation where males are highly competitive in the mating market, while females provide most of the parental care. However, there is a second equilibrium where males do most of the caring while the evolution of female choosiness is suppressed. Our simulations provide evidence that, in our model, the parental care pattern drives sexual selection and not the other way around. Fourth, our simulations suggest that (parental and mating) sex roles are evolutionarily labile. For most of the parameters considered, the model has two stable equilibria. Whenever this is the case, a simulation attains one of these equilibria for a long but limited period of time, followed by a rapid transition to the other equilibrium. Finally, our simulations shed fresh light on the 'Trivers effect'[12], which states that the sex with the highest pre-mating investment is predestined for doing most of the post-mating parental care. Although we disagree with Trivers' line of argumentation, most of our simulations recover this effect even under random mating conditions, demonstrating that it does not depend on factors such as sexual selection or uncertainty of paternity. Intriguingly, under some conditions (see Fig. 7 and Supplementary Fig. 9a), we observed the opposite outcome (that the sex with lower pre-mating investment is predestined for shouldering most of the post-mating

care). This exemplifies how difficult it is to disentangle the web of causal factors underlying the evolution of parental sex roles.

Why do our simulations lead to different conclusions than the earlier mathematical analyses of very similar model frameworks? Standard analytical models are typically based on simplifying assumptions in order to be tractable. It is well known that mathematical models only provide reliable evolutionary outcomes under restrictive conditions, such as weak selection[47], simple interactions across loci[48], uncorrelated mutations of similar effect sizes[49], and a simple structure of the genetic variance–covariance matrix[50]. Therefore, selection gradient-based plots like Fig. 2a should not be over-interpreted because it is not self-evident that evolution by natural selection proceeds in the direction of the selection gradient (the direction of the steepest ascent of the fitness landscape). Moreover, analytical approaches implicitly assume that the population is monomorphic (or that traits are distributed unimodally around the population average). In recent years, however, it is becoming increasingly clear that in the behavioural domain this assumption is not satisfied: in virtually all animals studied, individuals differ strongly and systematically in all kinds of behavioural tendencies[51–53] (including parental behaviour[54,55] and mating behaviour[56,57]), exhibiting polymorphic states. This is referred to as 'behavioural syndromes'[52] or 'animal personalities'[58]. Figures 3 and 4d, e show that such polymorphisms in parental strategies, within and between the sexes, are also to be expected in the evolution of sex roles. It has been argued before that polymorphisms can strongly affect the course of evolution (reviewed in refs. 59 and 60). Our simulations provide the insight that even a short-term polymorphism can have a long-lasting effect on the evolutionary outcome: the emergence of polymorphisms in care strategies is, in virtually all our simulations, the first step towards the evolution of sex role specialisation. Since this type of polymorphism is transient and thus rarely observed, its importance for explaining patterns that occur in nature may be underestimated.

Alternative stable strategies occur repeatedly in our simulations. We found that in situations with alternative stable equilibria, mating and parental strategies rapidly switch from one state to another when simulations are run for a sufficiently long time period. This is consistent with the findings of phylogenetic comparative studies, which also came to the conclusion that parental care patterns can change over the course of evolution, and that transitions from one care pattern to a different one have frequently taken place in a wide variety of animal taxonomic groups[9–11]. Relatively few studies address such switching behaviour. For example, it has been argued that evolutionary transitions between parental care patterns can be triggered by changes in fertilisation mode[61,62], or changes in life-history characteristics[43]. In our studies, however, these transitions were not driven by external changes, as environmental conditions were kept constant in all our simulations. This is less surprising than it may seem. In a stochastic dynamical system with alternative stable states, spontaneous transitions do regularly occur (e.g., ecological systems[63,64], the climate system[65], and physical systems[66], including the spontaneous reversal of polarity in the Earth magnetic field[67]). Interestingly, virtually all transitions in our simulations were preceded by the emergence of (transient) polymorphisms. This is, for example, illustrated by Fig. 4 and Supplementary Fig. 7. From these results, we conclude that the evolutionary lability of parental and mating patterns that is indicated by frequent transitions between patterns does not necessarily require an explanation in terms of changing environmental conditions or changing life-history features of the organisms.

We have shown that individual-based simulations can increase our understanding of the evolution of sex roles. However, we do not want to downplay the downsides of a simulation approach. Most importantly, it is often difficult to prove the robustness of the simulation results. Numerous parameter combinations need to be investigated, which is computationally demanding and time-consuming. Our

conclusions are based on tens of thousands of simulations. Even so, we initially missed part of Fig. 7, because we extrapolated the two extreme scenarios (very high and very low mortality in the pre-mating period, see Fig. 3 of ref. 44) and concluded that our simulations are in line with Trivers' hypothesis. We only discovered later that introducing intermediate mortality results in more sophisticated patterns. Similarly, we cannot rule out the possibility that some of our conclusions do not apply to unstudied parts of the parameter range. For example, the simulations reported here were all based on a relatively large value of the parameter $B$ ($B > 10$). It remains to be seen whether our results can be extrapolated to $B < 10$.

Despite these limitations of the simulation approach, our study demonstrates its various advantages. Simulations are easy to implement, without the necessity of performing complicated fitness calculations. Stochasticity, spatial structure, and environmental variation can easily be included in simulation models, in a variety of ways. Perhaps most importantly, individual interactions can be implemented in a more natural way than in analytical models[68]. Therefore, we believe the time has come to complement analytical sex role theory with evolutionary simulations that consider more complicated (and more realistic) scenarios, in which the sexes differ in life-history characteristics (creating a biased sex ratio), in which organisms make their decisions dependent on their own state and on environmental conditions, and in which 'good genes' and 'direct benefits' variants are included in sexual selection models[68,69].

## Methods

### Model structure

In line with the models of Kokko and Jennions[22] and Fromhage and Jennions[27], we consider a population with overlapping generations and a discrete-time structure. To be concrete, we assume that a time unit corresponds to one day. The population consists of females and males that, on each day, can be in one of the following states: juvenile, pre-mating, mate search, or caring. In each of the four states, there is a fixed mortality rate, which can be sex-specific. Unless stated otherwise, all mortalities were set to 0.001 day$^{-1}$. Therefore, the expected lifespan of an individual is 1000 days, a value that we consider a proxy for generation time. Offspring mortality is density-dependent, thus ensuring a limited population size. In our baseline scenario, population size fluctuates around 2000 females and 2000 males.

The complete life cycle of our model organisms is illustrated in Supplementary Fig. 1. Offspring that survive the period of parental care spend a fixed number of days (the maturation time) in the juvenile state. In all simulations reported, the maturation time of both sexes was equal to 20 days. After maturation, the surviving individuals enter the pre-mating state, corresponding to a condition where they prepare for mating (e.g., territory establishment; nest building; replenishment of gametes). After a fixed sex-specific number of days, the pre-mating state changes into the mate search state. Unless stated otherwise, the pre-mating period was set to zero, meaning that individuals move to the mate search state without delay. This case is illustrated in Fig. 1. Once in the mate search state, individuals seek for mating opportunities. In our baseline scenario, females and males mate at random, but we also consider a mate-choice scenario where females have a preference for certain male ornaments. On a given day, mating is modelled as follows: one by one, a female in the mate search state is selected at random. As long as there are still males in the mate search state, the female encounters one of these males at random. In the random mating scenario, such an encounter always results in mating; in the mate-choice scenario, the male can be rejected if its ornamentation does not fit to the preference of the female (see below). When mating does occur, both the male and the female immediately leave the mate search state and both enter the caring state. When a female-male encounter does not result in mating, both individuals stay in the mate search state, but they are no longer available for mating on

that day. Hence each individual in the mate search state can only have one encounter per day, and a female and a male both lose one day if their encounter does not result in mating. Mating will stop for the day when no more males in the mate search state are available and/or when all females in the mate search state have made their mating decisions. All remaining individuals stay in the mate search state, but they will only have a new mating opportunity on the following day.

Once a mating has occurred, the mated couple produces a clutch of offspring. Offspring survival strongly depends on the amount of parental care received. The female care duration $T_f$ and the male care duration $T_m$ are heritable traits that may differ between individuals. The evolution of $T_f$ and $T_m$ is the core subject of our study. We interpret $T_f$ and $T_m$ as the 'intended' cared duration: if one of the parents dies during the care period, this intended care duration is replaced by the actual care duration (the time from mating to death). To consider the possibility of synergy between the two parents, we assume that their total parental effort is given by $T_{tot} = T_f + T_m + \sigma T_f T_m$, where the 'synergy' parameter $\sigma$ is non-negative. Unless stated otherwise, we assume that $\sigma = 0$, meaning that each parent has an independent additive effect on total care. Offspring survival is proportional to $S(T_{tot}) = T_{tot}^2 / (T_{tot}^2 + B^2)$, an increasing sigmoidal function of total parental care. It is useful to give an interpretation of the parameter $B$. Consider the case of uniparental care of duration $T$. In our standard scenario where there is no pre-mating state, the expected number of matings of the caring parent is proportional to $1/T$, as the caring parent can immediately find a mating partner upon re-entering the mate search state. Hence, the expected lifetime reproductive success of the caring parent is proportional to $W(T) = S(T)/T = T/(T^2 + B^2)$, a function that is maximised for $T = B$. As a result, the parameter $B$ corresponds to the optimal care duration in case of uniparental care, which provides a useful benchmark expectation for the duration of care. Throughout, we consider the case of $B = 20$, i.e., our benchmark expectation corresponds to 20 days of care. When the care period $T_f$ (resp. $T_m$) has passed, the corresponding parent changes into the pre-mating state. When the longest-caring parent stops caring, the surviving offspring enter the juvenile state. As mentioned above, population size is regulated in our model by assuming that offspring survival is density-dependent: it is given by $S(T_{tot})/(1 + \gamma N)$, where $N$ is the current population size and the parameter $\gamma$ quantifies the degree of density dependence. This form of density regulation ensures that expected lifetime reproductive success (the fitness measure used by analytical approaches; see below) does indeed predict the course and outcome of evolution[70]. Our choice $\gamma = 0.003$ ensured relatively large populations (about 2000 females and 2000 males) with limited genetic drift and demographic stochasticity.

At the start of a new day, the survival of each individual was checked according to the individual's sex- and state-specific mortality. Non-survivors were removed from the population.

### Sexual selection

In part of our study, we consider a mate-choice scenario where females can evolve a preference $p$ for a male trait of size $s$, where $p$ and $s$ are both heritable traits. In line with Kokko and Johnstone[40], we assume that the probability that a female with preference $p$ that encounters a male with trait size $s$ will actually mate with this male is given by the logistic expression $(1 + \kappa \exp(\alpha(p - s)))^{-1}$ For all non-negative values of $p$, this expression increases with $s$ (hence all females have a preference for males with larger ornament sizes), and the rate of increase is positively related to $p$ (hence females with a large value of $p$ discriminate more strongly against males with a small trait size). The parameters $\kappa$ and $\alpha$ are scaling factors that affect the intensity of sexual selection. The mate-choice simulations shown are all based on the parameter values $\kappa = 0.02$ and $\alpha = 2$. For these parameters, an 'unattractive' male with $s = 0$ is accepted for mating with probability 0.98 by

a female with a preference value $p = 0$ (hence, $p = 0$ is almost indistinguishable from random mating) and with probability 0.48 by a female with preference value $p = 2$. We assume that male ornamentation is costly: each time step, the survival probability of a male with trait size $s$ is reduced by a percentage $\beta s^2$ where we chose $\beta = 10^{-6}$.

### Reproduction and inheritance

For simplicity, we consider a population of haploid individuals that may differ in their alleles at four gene loci. The $T_f$-locus and the $p$-locus are only expressed in females, and the $T_m$-locus and the $s$-locus are only expressed in males. The alleles at the $T_f$-locus and the $T_m$-locus determine the duration of maternal and paternal care, respectively. The allele at the $p$-locus determines the degree of female preference, while the allele at the $s$-locus determines the size of the male trait. In our baseline scenario (random mating), the $p$-locus and the $s$-allele are not expressed. Offspring inherit their alleles from their parents' subject to mutation. In a first step, the allele at each locus is drawn at random from one of its parents. Moreover, offspring sex is determined at random, with equal probability. In a second step, mutations could occur with probability $\mu = 0.005$ per locus. If a mutation occurs at the $T_f$-locus or the $T_m$-locus, the current allele is either increased or decreased by 1, with equal probability. This ensures that the parental care times $T_f$ and $T_m$ are natural numbers. If a mutation occurs at one of the other two loci, a small mutational step of size $\varepsilon$ was drawn from a Cauchy distribution (with location parameter 0 and scale parameter 0.01) and added to the current value of $p$ or $s$, respectively. We used the Cauchy distribution (rather than a normal distribution) because it allows for occasional larger step sizes. However, we limited mutational step sizes to a maximum value of $\varepsilon_{\max} = 0.05$.

### Initialisation and replication

In all simulations, the $p$- and the $s$-locus were initialised at $p = s = 0$. The $T_f$-locus and the $T_m$-locus were initialised at different values (leading to the different trajectories in Figs. 2b, 4b and 6a); each time, we started with a monomorphic population. For each parameter combination, we ran at least 100 replicate simulations. In all cases, the outcome was highly repeatable, allowing us to focus on one or two replicates. As partly documented in the Supplement, we also ran numerous simulations for model variants that differed from the baseline model in its parameter values (state- and sex-specific mortalities; the parameter $B$; cost of ornamentation $\beta$; density dependence $\gamma$; mutation rate $\mu$), the survival function $S(T_{tot})$, the mate-choice function, or the distribution of mutational step sizes. In all cases, we arrived at the same conclusions as reported in the manuscript. We therefore conclude that our results and conclusions are quite robust.

### Mathematical analysis

As a standard of comparison for our individual-based simulations, Fig. 2a shows the trajectories of the corresponding deterministic model, making use of the fitness gradient method described in Kokko and Jennions[22] and Fromhage and Jennions[27]. In a nutshell, this method calculates the selection gradient (indicating the strength and direction of selection) in males and females for each combination of parental care parameters $(T_f, T_m)$. This gradient points into the direction of the steepest ascend of the fitness landscape, where fitness is defined by expected lifetime reproductive success. Under the assumption that evolution will proceed in the direction of the selection gradient, evolutionary trajectories as in Figs. 2a and 4a are obtained. Our model is inspired by the model of Kokko and Jennions[22] and Fromhage and Jennions[27], but it differs from the former models in various respects. In the Supplement (Supplementary Figs. 10–12), we discuss these differences and demonstrate that our main results are also recovered for the earlier models, again indicating the robustness of our results and conclusions.

### Reporting summary

Further information on research design is available in the Nature Portfolio Reporting Summary linked to this article.

## Data availability

All simulated data that were used to generate figures in this study can be found at https://zenodo.org/record/8114131.

## Code availability

The C++ code for individual-based simulations (can be compiled using Visual Studio Community 2019 on Windows, XCode 15 on Mac, or the G++ Compiler on Linux), the R-script for data analysis and the Mathematica file for mathematical analysis are available for download from https://zenodo.org/record/8114131.

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

## Acknowledgements

We thank G.S. van Doorn, R. Scherrer, J. Komdeur, T. Székely and the MARM group at the University of Groningen for valuable discussion, comments, and suggestions. We are grateful to L. Fromhage for sharing technical details of the fitness gradient method and some computational resources with us. We appreciate the help of H. Hildenbrandt and M. Mosna with our programming. We thank the Centre for Information Technology of the University of Groningen for their support and for providing access to the Peregrine high performance computing cluster. X.L. was supported by a PhD fellowship of the Chinese Scholarship Council (NO. 201606380125). F.J.W. acknowledges funding from the European Research Council (ERC Advanced Grant No. 789240).

## Author contributions

X.L. and F.J.W. conceived the study and developed the model. X.L. performed the computational work and analysed the data. X.L. and F.J.W. interpreted the results and wrote the manuscript.

## Competing interests

The authors declare no competing interests.
