## [Peer Review File · Nature Communications]

Transient polymorphisms in parental care strategies drive divergence of sex rolesReviewers' Comments:

Reviewer #1:

Remarks to the Author:

This manuscript reports simulation results about the evolution of parental care. Although a similar setup has already been modelled mathematically, the present study's results are highly interesting and, in some ways, remarkably different from earlier results. The main difference is that, under conditions where earlier models predicted a neutral line of equilibria, the present model finds a strong tendency for sex role specialization. The authors do a good job of explaining how this divergence occurs via a transitory phase of within-sex polymorphism.

On the other hand, the manuscript also contains results which the authors have not been able to explain: as the duration of the pre-mating period increases, the sex experiencing this period initially becomes less but eventually more likely to specialize in parental care. This is an interesting pattern, but without identifying the underlying mechanism it is impossible to judge whether this mechanism is likely to operate in nature as well. This is not a fatal flaw, but, in my view, it is still a substantial drawback. I wonder if the authors have exhausted all reasonable options to get to the bottom of this problem. In particular, for reasons outlined below, I wonder how Fig. 7 would change if (i) the simulations started from a low level of paternal care and (ii) if the mutation rates were used to manipulate the trait-specific amounts of standing genetic variation.

Introducing a costly premating period (say, for females) should weaken selection, because more females will die without ever getting a chance to express their behaviour. This should affect both the mean and the variance of the female trait, with potentially counteracting effects on sex-role evolution. If the simulations start from a high initial level of parental care, then weaker selection should make females lag behind on the evolutionary path towards lower care. This will generate an asymmetry in sex-specific care levels, presumably predisposing females to end up providing most of the care. On the other hand, being exposed to weaker selection, the female trait should harbour more standing genetic variation. This might speed up the transition to within-sex polymorphism, perhaps predisposing females to end up providing little care. The details here are speculation, but I think these ideas could be tested.

Specific comments

The term "mating state" for as yet unmated individuals seems confusing. Perhaps use "available state" or "unmated state"?

L232: "tacitly" – you probably mean "tentatively"

L338: "Even though," – you probably mean "Even so,"

L352: I don't see a clear link between the preceding textbook characterization of simulations and the appeal for "a more nuanced sex role theory that explains how multiple mutually dependent factors interact with one another". The latter aspect has been studied quite successfully with mathematical models, e.g. McNamara & Wolf (2015), FJ16, Henshaw et al (2019). Arguably, these studies bring out more clearly than the present study what may happen when parental care coevolves with other traits which, in turn, affect the ability to provide parental care efficiently.

L404: The expected number of matings need not be proportional to $1/T$. This is only so if there are no delays between matings, such as might arise e.g. due to a pre-mating period or a biased OSR.

L411: "infant" – juvenile?

Legend to Figure S8. It's not clear to me in what sense this S function is "more symmetric"?

Legend to Figure S10. The emphasis here that *all* simulations resulted in the evolution of sex-biased care makes me wonder whether this outcome was specifically aimed for by using the Cauchy distribution instead of the more commonly used Normal distribution. If so, this approach could be seen as somewhat shady.

I choose to unblind myself as Lutz Fromhage.

Reviewer #2:

Remarks to the Author:

The authors use agent-based modeling (ABM) to address the question of the conflict between the sexes over the degree of parental care each provides. They use an existing mathematical modeling framework that is generally considered the current cutting edge. The results on parental care evolution of the ABM based on that framework, however, are quite different from the original model.

Comments

This manuscript is a major contribution to the literature on the theory of division of parental care between the sexes across a broad array of taxa. The authors first clearly review the current status of theory on the topic, which has been contentious. They propose the novel use of agent-based modeling to re-evaluate predictions that have been made in past theory. Then ABM is used judiciously by the authors to examine the question of how the division of parental care has evolved, which has been debated for decades. I believe that the methodology is appropriate and correct. The results strongly contradict earlier mathematical models in crucial ways, which will generate a lot of interest. The authors' results cast general question on whether prior mathematical models, because of their simplifications, produce reliable results in this area of evolutionary ecology.

Figure 2b. The trajectories in this figure of random mating are a little hard to read. It appears that all trajectories that start with a female-biased care converge on the high female care equilibrium, all that start with male-biased care converge to the high male care equilibrium, and those that start with equal care can randomly go either way. The authors have let the figure speak for itself, which is reasonable, as verbal interpretations of complex phenomena such as this can be vague and misleading. However, the authors might discuss the mechanisms a bit. The movement towards one or another alternative states must involve positive feedback that is eventually balanced by negative feedback. The fact that the ABM simulates individuals means that there can be competition of genotypes within the two populations. Starting with small mean values of both T_f and T_m , but with $T_f > T_m$, it would seem that female individuals that have T_f larger than the mean would produce more offspring than those with T_f less than the mean. The feedback due to more offspring would result in a larger mean T_f in the next generation. Because $S(T_{total})$ for low T_{total} is sensitive to increases in T_f , positive feedback should dominate until the costs of providing care leads to negative feedback on female-biased care stop the trajectory at an equilibrium point. I am not sure if this simple feedback is a sufficient explanation. I wonder if the authors have any thoughts on whether verbal explanation is possible for the results obtained by simulations.

Lines 189-202. The switching phenomenon is a nice emergent property of the ABM. As the authors note, switching between alternative stable states is not uncommon in models or in nature. They also note that it occurs in their model more frequently for low population numbers. However, the fact that it occurs even for large population numbers means that simple demographic stochasticity is not sufficient to trigger a switch between alternative states. There may be a random occurrence of a both relatively large fraction of the main care-givers that defect from care-giving and sufficient non-care-givers that defect from non-care-giving to set up some sort of amplifying feedback, in which both the females and males involved in this cluster of defectors are able to outcompete the others of their sex.

Figure 5 shows where in time these occurrences of self-reinforcing trends occur, which indicate that one sex seems to initiate the switch, but it would be interesting to take a closer look at exactly what distributions of genotypes and perhaps the sequences of matings that were taking place. Perhaps there is a way of making an early forecast that such a switch is likely to soon occur.

Figure 7. Unfortunately, I found this figure hard to understand. The key point made by the authors is: "Fig 7 shows that, actually, the situation is more complicated: if the pre-mating period is short and the mortality costs are at an intermediate level (light grey and dark grey curves), it is the sex with the lower pre-mating investment that evolves more frequently a high post-zygotic care level". I don't know why, but I am not able to interpret that from the figure. That may be my fault, but perhaps either a revised figure or more explanation would help.

Lines 203-225. The modeling here establishes a causal connection between parental care strategy and sexual selection. The evolution away from female-biased care was shown to drive female selection of males with ornaments as well as males towards more ornamentation. In nature this would seem reasonable that females that are the sole caregivers would choose to mate with males with the fittest males (e.g., best physiologically) and that ornamentation might be an indication that a male had those desirable characteristics. However, in the model it there does not seem to be any male attribute of genetic superiority associated with the ornamentation, so I don't understand the results of the causal connection. It is possible, though, that care-giving females have a preference to mate with ornamented males simply because her male offspring will be attractive to the next generation of females, but I am not sure whether that is the driving mechanism.

Line 60. Change 'and error-prone' to 'and are error-prone'

Response to Reviewers

Overall Response:

We very much appreciate the insightful comments and suggestions from reviewers. These comments are all very valuable and helpful in terms of improving the overall quality of our manuscript. We have read all comments thoroughly and attempted to address them all. The point-by-point responses are listed below, with the comments in blue and our responses in black. All changes to the manuscript have been traced.

Reviewer #1:

This manuscript reports simulation results about the evolution of parental care. Although a similar setup has already been modelled mathematically, the present study's results are highly interesting and, in some ways, remarkably different from earlier results. The main difference is that, under conditions where earlier models predicted a neutral line of equilibria, the present model finds a strong tendency for sex role specialization. The authors do a good job of explaining how this divergence occurs via a transitory phase of within-sex polymorphism.

We are grateful to the reviewer for the overall positive evaluation.

On the other hand, the manuscript also contains results which the authors have not been able to explain: as the duration of the pre-mating period increases, the sex experiencing this period initially becomes less but eventually more likely to specialize in parental care. This is an interesting pattern, but without identifying the underlying mechanism it is impossible to judge whether this mechanism is likely to operate in nature as well. This is not a fatal flaw, but, in my view, it is still a substantial drawback. I wonder if the authors have exhausted all reasonable options to get to the bottom of this problem. In particular, for reasons outlined below, I wonder how Fig. 7 would change if (i) the simulations started from a low level of paternal care and (ii) if the mutation rates were used to manipulate the trait-specific amounts of standing genetic variation. (double step size of male mutations)

Introducing a costly premating period (say, for females) should weaken selection, because more females will die without ever getting a chance to express their behaviour. This should affect both the mean and the variance of the female trait, with potentially counteracting effects on sex-role evolution. If the simulations start from a high initial level of parental care, then weaker selection should make females lag behind on the evolutionary path towards lower care. This will generate an asymmetry in sex-specific care levels, presumably predisposing females to end up providing most of the care. On the other hand, being exposed to weaker selection, the female trait should harbour more standing genetic variation. This might speed up the transition to within-sex polymorphism, perhaps predisposing females to end up providing little care. The details here are speculation, but I think these ideas could be tested.

We fully understand the reviewer's wish for further clarification. Actually, we devoted a lot of time and simulation effort to finding a convincing explanation for the intriguing pattern in Fig. 7. Whenever we thought to have found such an explanation (e.g. the Sutherland mechanism reported in the manuscript), subsequent analyses proved this explanation to be wrong. Inspired by the reviewer's comments, we ran additional simulations. We checked whether the sex with the longer pre-mating period "lags behind" in its approach to the "branching point" ($T_f = T_m = 5$), but even in simulations with a very long pre-mating period, such an effect was hardly noticeable. This is confirmed by the **new**

Supplementary Fig. 9a, which demonstrates that a virtually identical pattern as in Fig. 7 emerges when the population is started at a low egalitarian care level. After having checked various other initial conditions, we are reasonably sure that the pattern in Fig. 7 is robust (for our model), at least if the population is initialised at egalitarian care.

Inspired by the comments of Reviewer 2, we also considered asymmetric initial conditions in more detail. As illustrated in the **new Supplementary Fig. 9b**, such initial asymmetries have a strong effect on the outcome. We elaborate on this in **the new Supplementary Fig. 3**. We also run simulations with sex-differential mutation rates. Such differences may indeed tip the balance regarding which parent is more likely to do the caring, but the effect is not straightforward. If, for example, females have a longer pre-mating period and a higher mutation rate (0.0075 vs 0.0025), 94 of 100 simulations ended up in strongly male-biased care when the population was started at $T_f = T_m = 20$, while 98 of 100 simulations ended up in strongly female-biased care when the starting point was $T_f = T_m = 5$. We are able to explain these differences, but as they do not help to explain the pattern in Fig. 7, we decided not to include these results in the current manuscript.

We still do not have a fully satisfactory explanation for the patterns in Fig. 7 (in particular for the non-monotonicity of the two grey curves corresponding to intermediate pre-mating mortality). But we have now expanded our treatment on this, providing more explanation than before (**lines 280-306**).

Specific comments

The term “mating state” for as yet unmated individuals seems confusing. Perhaps use “available state” or “unmated state”?

Thanks for spotting this. The term "mating state" has been replaced with "mate search state."

L232: “tacitly” – you probably mean “tentatively”

Corrected.

L338: “Even though,” – you probably mean “Even so,”

Corrected.

L352: I don't see a clear link between the preceding textbook characterization of simulations and the appeal for “a more nuanced sex role theory that explains how multiple mutually dependent factors interact with one another”. The latter aspect has been studied quite successfully with mathematical models, e.g. McNamara & Wolf (2015), FJ16, Henshaw et al (2019). Arguably, these studies bring out more clearly than the present study what may happen when parental care coevolves with other traits which, in turn, affect the ability to provide parental care efficiently.

We agree and rephrased this part (now **lines 391-396**).

L404: The expected number of matings need not be proportional to $1/T$. This is only so if there are no delays between matings, such as might arise e.g. due to a pre-mating period or a biased OSR.

We agree and have clarified this (**lines 441-446**).

L411: “infant” – juvenile?

Changed.

Legend to Figure S8. It's not clear to me in what sense this S function is “more symmetric”?

We agree and deleted the “more symmetric”.

Legend to Figure S10. The emphasis here that *all* simulations resulted in the evolution of sex-biased care makes me wonder whether this outcome was specifically aimed for by using the Cauchy distribution instead of the more commonly used Normal distribution. If so, this approach could be seen as somewhat shady.

This is not a problem, as the Cauchy distribution was not used for determining the mutational step size at the loci determining the care duration T_f and T_m . The Cauchy distribution was only used for the female preference locus and the male ornament locus in the sexual-selection scenarios in Fig. 6 and Supplementary Fig. 8. To check whether the mutational distribution matters in these cases, we re-run all corresponding simulations, but now with a normally distributed mutational step size (with a mean of 0.0 and a standard deviation of 0.1). We obtained the same evolutionary outcomes as reported for the Cauchy distribution and conclude that our results are robust in this respect.

Reviewer #2:

The authors use agent-based modeling (ABM) to address the question of the conflict between the sexes over the degree of parental care each provides. They use an existing mathematical modeling framework that is generally considered the current cutting edge. The results on parental care evolution of the ABM based on that framework, however, are quite different from the original model.

Comments

This manuscript is a major contribution to the literature on the theory of division of parental care between the sexes across a broad array of taxa. The authors first clearly review the current status of theory on the topic, which has been contentious. They propose the novel use of agent-based modeling to re-evaluate predictions that have been made in past theory. Then ABM is used judiciously by the authors to examine the question of how the division of parental care has evolved, which has been debated for decades. I believe that the methodology is appropriate and correct. The results strongly contradict earlier mathematical models in crucial ways, which will generate a lot of interest. The authors' results cast general question on whether prior mathematical models, because of their simplifications, produce reliable results in this area of evolutionary ecology.

We very much appreciate the reviewer for this positive evaluation of our study.

Figure 2b. The trajectories in this figure of random mating are a little hard to read. It appears that all trajectories that start with a female-biased care converge on the high female care equilibrium, all that start with male-biased care converge to the high male care equilibrium, and those that start with equal care can randomly go either way. The authors have let the figure speak for itself, which is reasonable, as verbal interpretations of complex phenomena such as this can be vague and misleading. However, the authors might discuss the mechanisms a bit. The movement towards one or another alternative states must involve positive feedback that is eventually balanced by negative feedback. The fact that the ABM simulates individuals means that there can be competition of genotypes within the two populations. Starting with small mean values of both T_f and T_m , but with $T_f > T_m$, it would seem that female individuals that have T_f larger than the mean would produce more offspring than those with T_f less than the mean. The feedback due to more offspring would result is a larger mean T_f in the next generation. Because $S(T_{total})$ for low T_{total} is sensitive to increases in T_f , positive feedback should dominate until the costs of providing care leads to negative feedback on female-biased care stop the trajectory at an equilibrium point. I am not sure if this simple feedback is a sufficient explanation. I wonder if the authors have any thoughts on whether verbal explanation is possible for the results obtained by simulations.

We are grateful for these comments. Most of our more detailed simulations used high-level egalitarian care ($T_f = T_m = 20$) as their point of departure. Stimulated by the comments of Reviewer 1, we now also considered low-level egalitarian care ($T_f = T_m = 5$) as initial condition (see the **new Supplementary Fig. 9a**). We now also discuss the non-egalitarian case ($T_f \neq T_m$) in more detail. We added a new paragraph on this to the main text (**lines 151-160**), which is further illustrated and explained in the **new Supplementary Fig. 3**. The **new Supplementary Fig. 9b** also addresses the importance of initial asymmetries on the evolutionary outcome. Moreover, to make Fig. 2b easier to understand, we explicitly pointed out that the time trajectories shown in Fig. 2c and 2d correspond to the simulations depicted as yellow-coloured lines in Fig. 2b, and that one of the time trajectories in **Supplementary Fig. 3** corresponds to a simulation depicted as a blue-coloured line in Fig. 2b.

Lines 189-202. The switching phenomenon is a nice emergent property of the ABM. As the authors note, switching between alternative stable states is not uncommon in models or in nature. They also

note that it occurs in their model more frequently for low population numbers. However, the fact that it occurs even for large population numbers means that simple demographic stochasticity is not sufficient to trigger a switch between alternative states. There may be a random occurrence of a both relatively large fraction of the main care-givers that defect from care-giving and sufficient non-care-givers that defect from non-care-giving to set up some sort of amplifying feedback, in which both the females and males involved in this cluster of defectors are able to outcompete the others of their sex. Figure 5 shows where in time these occurrences of self-reinforcing trends occur, which indicate that one sex seems to initiate the switch, but it would be interesting to take a closer look at exactly what distributions of genotypes and perhaps the sequences of matings that were taking place. Perhaps there is a way of making an early forecast that such a switch is likely to soon occur.

We fully agree that additional explanations would be helpful. To this end, we studied the transitions between alternative states in hundreds of simulations in considerable detail. Our conclusions are summarised in a new paragraph (**lines 214 and 222**), which is supported by the **new Supplementary Fig. 7**. In a nutshell, such transitions are all preceded by the convergence of the system to the 'evolutionary branching point' $T_f = T_m = 5$, from where the system returns to one of the alternative stable states via transient polymorphisms. As shown in Fig. 5 and **new Supplementary Fig. 7**, it is possible that the system returns to the previous stable state (actually, this is the most likely outcome), but it can also switch to the alternative stable state, via the route explained in detail in Fig. 3. There is indeed a clear 'warning signal' for the potential switch to another attractor: whenever the care level of the less-caring sex approached the level of 5, the more-caring sex rapidly evolved to this level as well. Relatively small fluctuations in the care level of the less-caring sex are therefore sufficient to initiate attractor switching.

Figure 7. Unfortunately, I found this figure hard to understand. The key point made by the authors is: "Fig 7 shows that, actually, the situation is more complicated: if the pre-mating period is short and the mortality costs are at an intermediate level (light grey and dark grey curves), it is the sex with the lower pre-mating investment that evolves more frequently a high post-zygotic care level". I don't know why, but I am not able to interpret that from the figure. That may be my fault, but perhaps either a revised figure or more explanation would help.

We have now explained Fig. 7 in more detail. **Lines 280-306** have been largely rewritten, and a **new Supplementary Fig. 9** has been added to the manuscript. We hope that both aspects clarify the figure (and our problems in finding a satisfying explanation for the two grey curves).

Lines 203-225. The modeling here establishes a causal connection between parental care strategy and sexual selection. The evolution away from female-biased care was shown to drive female selection of males with ornaments as well as males towards more ornamentation. In nature this would seem reasonable that females that are the sole caregivers would choose to mate with males with the fittest males (e.g., best physiologically) and that ornamentation might be an indication that a male had those desirable characteristics. However, in the model it there does not seem to be any male attribute of genetic superiority associated with the ornamentation, so I don't understand the results of the causal connection. It is possible, though, that care-giving females have a preference to mate with ornamented males simply because her male offspring will be attractive to the next generation of females, but I am not sure whether that is the driving mechanism.

The reviewer is completely correct that our results are based on the Fisherian model, which means that the production of attractive sons is the only benefit of female choice. To clarify this, we have added the term 'sexy-son model' to the model description (**line 233**). We fully agree with the reviewer

that it remains to be seen whether other benefits of female choice (e.g. good genes or direct benefits) lead to similar conclusions. We therefore avoid overinterpreting the conclusions “of this simple model” (lines 252-254) and stress the necessity of additional research in the final sentence of our manuscript (lines 391-396).

Line 60. Change ‘and error-prone’ to ‘and are error-prone’

Changed.

Additional corrections: Throughout the manuscript, we corrected various errors and typos. Most importantly, we corrected the number of generations in Fig. 7 (100,000 instead of 100,000,000).

Reviewers' Comments:

Reviewer #1:

Remarks to the Author:

I am satisfied with the revisions and congratulate the authors on a cool paper. Some minor comments are below.

Best wishes, Lutz Fromhage

L 89 "In other words, it is possible to keep any type of care strategy in place as the initial conditions, including female-only care, egalitarian biparental care, male-only care, and everything in between."
- Did you mean "at the initial conditions"?

L126 Since each individual lives only once, it's not quite clear what it means for an individual to be selected to care more (or less) than it currently does. Perhaps rephrase as "individuals that provide a relatively high level of care could benefit from providing even more care".

Legend to Fig 7: "Mortality in the pre-mating period was ...(2) the same as in the mate search phase"
Please clarify numerically what "the same" means here.

Legend to Fig S7: "shows rapid switches from one equilibrium to the other from long-term scale" -
Perhaps revise as "frequent switches"... "on a long time scale". (That something occurs rapidly on a long time scale is an oxymoron.)

Legend to Fig S9: "'Trivers effect' was completely recovered when mortality was 0.005. When the mortality in the pre-mating period was 0.005 (black dots and line), Trivers effect' was completely recovered:". I suggest removing this redundancy. Also, I would write "*the* Trivers effect" (with article).

Legend to Fig S9: "but with a much higher probability of female care equilibrium evolving" – perhaps revise along the lines of "but with a much higher probability of care by the sex that provided more care initially."

Fig S9. Compared to Fig 7, the light and dark grey lines seem to be switched. Is this an error?

Reviewer #2:

Remarks to the Author:

The authors use an agent-based model to address the question of the conflict between the sexes over the degree of parental care each provides. They use an existing mathematical modeling framework that is generally considered the current cutting edge. The results on parental care evolution of the ABM based on that framework, however, are quite different from the original model.

Comments

The authors have responded to the first set of reviews with additional studies of their model that help to address the original comments. An important improvement in the revised manuscript is the authors' showing that the pattern of parental care shown in Figure 7 is preserved for different initial conditions than those in Figure 7. Also, the authors provide a mechanism for the parental care for the case when the pre-mating period does not affect life expectancy (Lines 280-286). That is, the partner with shorter pre-mating period in the mate searching phase, during which there is mortality, has to wait for a long period for members of the opposite sex to come into the mate searching phase. That higher mortality leads, according to Long (2022), to a bias towards care by the sex with lower mortality.

I still have a question concerning that interpretation. Above, in Lines 269-274, it is noted that the sex with longer pre-mating period, especially when mortality is much higher than in the mate searching period, will have the higher investment in post-zygotic parental care. That is reasonable, as the potential for future mating is diminished by the high mortality, which motivates devoting care to the present clutch. But it seems to me that the interpretations of these two situations is not quite consistent. In the case of mortality during the pre-mating phase, lower mortality is associated with a greater investment in post-zygotic care, whereas in the case where there is no mortality in that phase, higher mortality (in the mate searching phase) is associated with greater parental care. Some further clarification would be helpful for me.

Other than that possible issue, I don't have further questions about the results of this manuscript. The mechanism behind Figure 2b, which is a consequence of transient polymorphisms emerging in the populations of the sexes, seems reasonable. The formation of polymorphisms in populations of individuals, which can amplify in time due to some fitness tradeoffs, occurs in agent-based models of population in other contexts. This possibility is an important difference from deterministic models. This result, along with the other results, such as switching of parental roles in evolutionary time, make this manuscript very important to this key topic in evolutionary theory.

Response to Reviewers

Overall Response:

We would like to thank both reviewers once again for their time and consideration. We have addressed all additional comments. The point-by-point responses are listed below, with the comments in blue and our responses in black. All changes to the manuscript have been highlighted.

Reviewer #1:

I am satisfied with the revisions and congratulate the authors on a cool paper. Some minor comments are below.

Best wishes, Lutz Fromhage

Thank you again for your generally positive feedback.

L 89 "In other words, it is possible to keep any type of care strategy in place as the initial conditions, including female-only care, egalitarian biparental care, male-only care, and everything in between." - Did you mean "at the initial conditions"?

We fully agree with the reviewer that the original text is not clear. We have rephrased this part (**lines 98-101**).

L126 Since each individual lives only once, it's not quite clear what it means for an individual to be selected to care more (or less) than it currently does. Perhaps rephrase as "individuals that provide a relatively high level of care could benefit from providing even more care".

Thanks for pointing out this. **Lines 126-128** have been rephrased.

Legend to Fig 7: "Mortality in the pre-mating period was ...(2) the same as in the mate search phase" Please clarify numerically what "the same" means here.

The mortality in the pre-mating period has been clarified numerically for all cases in the legend of Fig. 7.

Legend to Fig S7: "shows rapid switches from one equilibrium to the other from long-term scale" - Perhaps revise as "frequent switches"... "on a long time scale". (That something occurs rapidly on a long time scale is an oxymoron.)

We have changed the wording and removed the reference to a long-term scale in the legend of Fig. S7 (as Figs 5 and S7 have the same time scale).

Legend to Fig S9: "'Trivers effect' was completely recovered when mortality was 0.005. When the mortality in the pre-mating period was 0.005 (black dots and line), Trivers effect' was completely recovered:". I suggest removing this redundancy. Also, I would write "*the* Trivers effect" (with article).

Legend to Fig S9: “but with a much higher probability of female care equilibrium evolving” – perhaps revise along the lines of “but with a much higher probability of care by the sex that provided more care initially.”

Thanks for spotting these. We have streamlined the whole legend of Fig. S9 and incorporated all of the suggestions provided by the reviewer.

Fig S9. Compared to Fig 7, the light and dark grey lines seem to be switched. Is this an error?

Thank you for noticing this mistake. After double-checking the data, it indeed turned out that we had accidentally switched the two colours in Fig. S9a. The figure has now been updated.

Reviewer #2:

The authors use an agent-based model to address the question of the conflict between the sexes over the degree of parental care each provides. They use an existing mathematical modeling framework that is generally considered the current cutting edge. The results on parental care evolution of the ABM based on that framework, however, are quite different from the original model.

Comments

The authors have responded to the first set of reviews with additional studies of their model that help to address the original comments. An important improvement in the revised manuscript is the authors' showing that the pattern of parental care shown in Figure 7 is preserved for different initial conditions than those in Figure 7. Also, the authors provide a mechanism for the parental care for the case when the pre-mating period does not affect life expectancy (Lines 280-286). That is, the partner with shorter pre-mating period in the mate searching phase, during which there is mortality, has to wait for a long period for members of the opposite sex to come into the mate searching phase. That higher mortality leads, according to Long (2022), to a bias towards care by the sex with lower mortality.

I still have a question concerning that interpretation. Above, in Lines 269-274, it is noted that the sex with longer pre-mating period, especially when mortality is much higher than in the mate searching period, will have the higher investment in post-zygotic parental care. That is reasonable, as the potential for future mating is diminished by the high mortality, which motivates devoting care to the present clutch. But it seems to me that the interpretations of these two situations is not quite consistent. In the case of mortality during the pre-mating phase, lower mortality is associated with a greater investment in post-zygotic care, whereas in the case where there is no mortality in that phase, higher mortality (in the mate searching phase) is associated with greater parental care. Some further clarification would be helpful for me.

We are sorry for not explaining why a lower mortality can be associated with a higher level of parental care. We have systematically explored how sex differences in mortality rates at the mating stage affect the evolution of parental care patterns in Long (2022, Chapter 4). Similar to our white dots and white lines shown in Fig.7 and Supplementary Fig. 9a, we found that the sex with lower mortality at the mating stage was typically selected to provide more parental care, but that the opposite outcome (that the sex with higher mortality does most of the caring at the equilibrium) also evolved in a smaller proportion of simulations (with the same parameter setting). The outcomes can be explained by two similar lines of reasoning that lead to opposing conclusions. The first line of reasoning is based on biased sex ratios and explains why a lower mortality is associated with a higher level of parental care. The argument goes like this: the sex with the lower mortality is overrepresented in the population. According to Fisher (1930), each offspring of diploid sexually reproducing organisms has one father and one mother. As a result, the total number of offspring produced by all males must be equal

to the total number of offspring produced by all females. As a consequence, each member of the minority sex (here: the sex with higher mortality) will, on average, produce more offspring than a member of the majority sex (here: the sex with lower mortality). In other words, the members of the majority sex have a lower *per capita* reproductive success. In our model, this also means that the expected number of future matings is lower for the majority sex than for the minority sex. In the trade-off between current and future reproduction, the members of the majority sex should therefore place (relatively) more emphasis on the current brood than the members of the minority sex. As a consequence, the majority sex that has lower mortality is more strongly selected to provide care. The second line of reasoning makes use of similar life-history considerations and explains why a higher mortality is associated with a higher level of parental care. The argument, which has been described in the manuscript and by the reviewer, is as follows: the sex with higher mortality has a shorter life expectancy and consequently a lower potential for future reproduction, this sex is therefore more strongly selected to invest in the current brood, leading to a parental sex bias toward the sex with higher mortality. In a situation where two intricately interwoven mechanisms lead to contrasting results, one of these mechanisms may play a more decisive role in determining the evolutionary outcomes, depending on the parameter setting. For instance, in Fig.7 and Supplementary Fig. S9a, the first mechanism is more essential in influencing care strategies when pre-mating mortality is zero and mating mortality is 0.001 (white dots and white lines), while the second mechanism is much more vital in determining the evolutionary outcomes when pre-mating mortality is 0.005 and mating mortality is 0.001 (black dots and black lines). We have expanded and modified the main text to explain Fig. 7 in more detail based on the two reasonings that lead to contrasting conclusions (**lines 281-296; lines 302-304; lines 307-311**). We hope that Fig. 7 is now more clarified.

Reference:

Fisher, R.A. 1930. *The genetical theory of natural selection*. Oxford University Press.

Other than that possible issue, I don't have further questions about the results of this manuscript. The mechanism behind Figure 2b, which is a consequence of transient polymorphisms emerging in the populations of the sexes, seems reasonable. The formation of polymorphisms in populations of individuals, which can amplify in time due to some fitness tradeoffs, occurs in agent-based models of population in other contexts. This possibility is an important difference from deterministic models. This result, along with the other results, such as switching of parental roles in evolutionary time, make this manuscript very important to this key topic in evolutionary theory.

Once again, many thanks for your comments and overall positive feedback!

Reviewers' Comments:

Reviewer #2:

Remarks to the Author:

I have read the revised version of the manuscript and the authors' responses to the comments. I am satisfied with the manuscript as it stands.